# Sex Differences in Adiposity and Cardiovascular Diseases

**DOI:** 10.3390/ijms23169338

**Published:** 2022-08-19

**Authors:** Haoyun Li, Daniels Konja, Luyao Wang, Yu Wang

**Affiliations:** The State Key Laboratory of Pharmaceutical Biotechnology, Department of Pharmacology and Pharmacy, The University of Hong Kong, Hong Kong SAR, China

**Keywords:** obesity, adipose tissue, sexual dimorphism, body fat distribution, cardiovascular diseases

## Abstract

Body fat distribution is a well-established predictor of adverse medical outcomes, independent of overall adiposity. Studying body fat distribution sheds insights into the causes of obesity and provides valuable information about the development of various comorbidities. Compared to total adiposity, body fat distribution is more closely associated with risks of cardiovascular diseases. The present review specifically focuses on the sexual dimorphism in body fat distribution, the biological clues, as well as the genetic traits that are distinct from overall obesity. Understanding the sex determinations on body fat distribution and adiposity will aid in the improvement of the prevention and treatment of cardiovascular diseases (CVD).

## 1. Introduction

Of the world’s population, one-third are currently overweight or obese (https://www.worldometers.info/obesity/, accessed on 1 July 2022). The rapid increase in obesity is threatening public health globally, including in China [1], where the prevalence has risen from approximately 0 to 16.4% (1982–2019) over the past ~40 years [2]. The pandemic of obesity has greatly burdened individuals, society, and the healthcare system. Obesity is responsible for approximately five million premature deaths each year and represents an independent risk factor for cardiovascular disease, the leading cause of global mortality and a major contributor to the disability [3,4]. The global prevalence of obesity in women is higher than in men. In 2020, the overall global obesity rate for women was estimated at 25% (vs. 17% for men), of which 54 million (vs. 22 million for men) are severe (Class III) BMI ≥ 40 kg/m^2^. By 2030, this number of women could be as high as 30% (vs. 20% for men) and 77 million (vs. 34 million) being severely obese [5]. Under obese conditions, the dysfunctional adipose tissue contributes to various pathologies in the cardiovascular system in a sex-dependent manner [6,7,8]. There is increasing interest in the pathophysiological differences between males and females in the incidence and consequence of obesity [4,9]. The present review aims to summarise the current knowledge on the sex differences related to adiposity and associated cardiovascular complications in humans.

## 2. Adipose Tissue: Classification, Distribution, and Function

Adipose tissues, also known as body fats, are energy-processing endocrine organs that are classically classified by their functions or anatomical distributions. Functionally, while energy-storing white adipose tissues (WAT) are distributed in almost every part of the human body, thermogenic-controlling brown adipose tissues (BAT) are mainly located in the interscapular and mediastinal regions with rich nerves and blood vessels [10,11,12]. Under the condition of an increasing energy intake, excessive triglycerides deposited in WAT lead to obesity. In contrast, BAT oxidises glucose and lipids through uncoupled mitochondrial respiration to generate heat, thus dissipating energy via adaptive heat production [13,14]. Anatomically, adipose tissues are classified as subcutaneous adipose tissues (SAT) making up over 80% of total fat in the body [15], and visceral adipose tissues (VAT) surrounding the different thoracic and abdominal organs. WAT surrounding the heart comprises the epicardial (ECAT) and pericardial adipose tissue (PCAT) [16]. The abdominal VAT, including the omental, mesenteric, and retroperitoneal fat depots, are highly metabolically active. Most blood vessels are surrounded by perivascular adipose tissues (PVAT). Depending on the anatomical positions, the cellular compositions and the properties of PVAT are different. For example, PVAT associated with the thoracic aorta resembles BAT, whereas those surrounding the abdominal aorta exhibit similarities with WAT [17].

Adipose tissue is not only an energy source, but also the largest endocrine organ in the body [18,19]. The protein factors secreted from adipose tissue are collectively referred to as adipokines. Emerging evidence suggests that adipose tissue has more colours. Beige adipocytes are a distinct type of WAT sharing similarities with the classic cells in BAT. Brown adipocytes and myocytes, which are derived from a Myf5-expressing cell lineage, exhibit similar developmental origins [20,21]. The beige adipocytes originate from different and heterogeneous populations of cell lineages and are characteristic of both white and brown fat cells [22]. Pink adipose tissues (PAT) are sex specific. During pregnancy, the female SAT of the mammary gland begins to transform into a reservoir as PAT, which gradually replaces the WAT during the lactation period. PAT turns into WAT again when breastfeeding ends [23,24]. The whole process is referred to as alveolarogenesis, which involves the development of the lobule-alveolar gland structure to produce milk [23,24]. PAT secretes leptin and adiponectin that act to prevent neonatal obesity [25,26]. The yellow adipocytes are related to those of the marrow adipose tissues (MAT) in bone. MAT accounts for more than 10% of the total fat mass in healthy lean people [27,28]. Similar to WAT, MAT also acts as a large endocrine organ that secretes leptin and adiponectin [28,29], which increase or decrease under pathological conditions such as osteoporosis [30,31], diabetes [32], and obesity [33].

Depending on the anatomical locations, adipose tissue depots show different metabolic and endocrine properties. The propensity to generate new adipocytes in different adipose depots varies, thus their expansions are intrinsically different leading to a diversified cellular composition, function, and cardiometabolic consequences [34]. Adipose-derived factors, including adipokines, are key mediators of the alterations in body fat composition with age [35,36]. Different fat depots produce a distinct profile of mediators, which is affected by age and pathophysiological conditions. For example, the VAT expresses a greater amount of inflammatory adipokines [17,37,38]. Even in the same individual, the *ob* (obese) mRNA level in the adipose tissue varies from region to region [39]. As a result, the production of leptin as well as other inflammatory cytokines such as angiotensinogen, interleukin 6 (IL-6), and plasmin activator inhibitor 1 from SAT and VAT are different [40]. Leptin produced in the SAT is closely related to the circulating concentration [41,42]. Overall, the heterogeneity in the distribution and functions of adipose tissues exerts different effects on body fat distribution (Figure 1).

## 3. Sex Dimorphism in Body Fat Distribution

Sex is defined as the biological feature of males and females determined by genetics, regardless of social or environmental influences. The XX (female) or XY (male) chromosomes make individuals’ distinct sex from each other genetically and physiologically. Sex dimorphism, which refers to the characteristic differences between males and females in a species, helps clinicians and researchers classify, treat, and offer the prognosis of diseases differently. Over the last decade, sex-specific medicine has been drawing great attention as one of the first and foremost advancements of personalised medicine. To date, sex dimorphism has been intensively investigated in obesity, cancers, neurodegenerative disorders, and cardiovascular, bone, and infectious diseases, as well as pain management. The sex disparities in different pathologies, with the development of sex-omics technology, particularly sheds light on personalised management in chronic and severe diseases [43].

Adiposity refers to the distribution of body fat while obesity is a more measurable parameter, emphasising the stratification metrics related to the BMI (ratio of weight to the square of height) and waist circumference [44]. Sexual dimorphism of body fat distribution is subtle in the early stages of life, more distinct in adolescence, and strongly present throughout adult life, but attenuated later in life [45,46,47,48,49]. Of those with the same BMI and similar age, women have a significantly higher amount of adipose tissue deposition, especially the lower extremity fat, than men [50]. By contrast, men often develop central obesity with an increased fat deposition around the abdomen [51]. With advancing age, fat mass increases and peaks around the age of 60–79 years, later in women than men [52]. Age-associated changes in body composition are manifested not only by an increase in VAT, a decrease in SAT, and an accumulation of ectopic fat, but also by a significant reduction in the lean mass [53,54,55,56]. With age, the muscle loses its mass, strength, and physical functionality, leading to a high-risk geriatric syndrome known as sarcopenic obesity (SO), which contributes to various medical complications [57]. SO shows the sex variation and is more prevalent in elderly women [58]. However, the epidemiological findings are heterogeneous due largely to the lack of consensus on a standard definition of SO [59]. The prevalence of SO ranges from 4.4% to 84.0% in men and 3.6% to 94.0% in women when assessed with dual-energy X-ray absorptiometry [60]. In Europe and the US, the prevalence of SO is greater in men than women when using the appendicular lean muscle mass divided by squared height (ALM/h^2^) to define SO [61,62,63,64,65]. An opposite conclusion is drawn by a study in Korea using ALM/weight (%) as the criterion [66]. A cross-sectional study from China shows men were more likely than women to have sarcopenia and SO, as assessed by the Asian Working Group for Sarcopenia (AWGS) [67]. Women with SO may have higher glucose, while men with SO are more likely to develop osteoporosis and dyslipidaemia [68].

Body fat distribution is modulated by sex hormones and their receptors [69,70,71]. For example, in women, augmented VAT changes the body shape and composition towards a more android type after menopause [69]. The phenomenon is due at least partly to the withdrawal of estrogen levels, which regulate the sexually dimorphic expression of genes involved in adipose tissue development, distribution, and function [72,73,74]. Sex-specific hormonal factors play an important role in the development of SO. In women, the decrease in estrogen levels after menopause leads to an increase in adiposity and a change in the fat distribution pattern, with a shift from subcutaneous to visceral deposits and muscle tissue [75]. In older men, the development of SO is more strongly associated with a decrease in the total testosterone levels, which causes a reduction in both muscle mass and strength [76]. The expression of sex hormone receptors also affects the distribution pattern of adipose depots. Sex hormone-related receptors are differentially expressed in SAT and VAT [77]. The expression levels of estrogen and progesterone receptors are high in SAT, whilst VAT show an increased amount of androgen receptors [78]. Estrogen acts as an antagonist to decrease the expression of androgen receptors [79]. Low total testosterone can also lead to visceral obesity [80]. The decrease in total and bioavailable testosterone is a more direct predictor of VAT accumulation and cardiovascular risk than the decrease in estradiol levels [81]. The sex hormones interact with transcription factors to regulate gene activity in a sex-dependent manner [69,82]. However, animal studies do not support the correlations between circulating sex hormones and obesity-related genes [83].

Intermuscular adipose tissue (IMAT) has been recognised as an independent fat depot in assessing insulin sensitivity, lipid and lipoprotein metabolism, and predicting cardiovascular risk [84,85,86]. Men with overweight and obesity have significantly higher neck IMAT accumulation as an ectopic fat [87]. The ratio between subcutaneous and intramuscular adipose tissue (SAT/(SAT + IMAT) is significantly associated with serum adiponectin levels in both men and women, but more strongly in the latter, while the correlations with SAT or IMAT alone are not significant for both sexes [88]. The different distribution of adipose tissue affects body shape, but not necessarily the overall BMI in women and men. Under thermoneutral conditions, women exhibit more BAT mass and greater thermogenic responses than men [89,90,91]. However, the sex differences in BAT diminish with age or in cold conditions [92,93,94,95,96,97]. Compared with men, PET-CT can identify more UCP1-immunopositive regions, represented as BAT, and higher ^18^F-fluorodeoxyglucose (^18^F-FDG) uptake activity in the area extending from the neck to the chest of women [98]. Men display a decreased response to cold exposure due to the lower mitochondrial function [99]. In addition, ageing in men induces a faster functional decline of BAT activity than in women [100]. Fat deposition of the tongue is higher in men than women and associated with decreased upper airway patency [101]. Compared to women, there is a significantly higher amount of PCAT in the men’s [102]. On the contrary, the ECAT volume is significantly increased in middle-aged and older Japanese women [103].

## 4. Sex Dimorphism and Obesity-Related Cardiovascular Abnormalities

Men and women are both susceptible to obesity but differ in the health consequences, due largely to the different body fat distribution [104,105,106]. CVD represents the most important cause of mortality and morbidity around the world, exhibiting a high heterogeneity in the epidemiology and management between male and female patients. The incidence of CVD is generally higher in men than in women [107], as abundant SAT in the lower-hip extremity region protects women from cardiovascular abnormalities [54,55,108], whereas a larger mass of VAT around abdominal organs in men contributes to the risk of premature atherosclerosis and acute coronary syndromes by activating pro-inflammatory factors [109]. However, studies have shown that obesity could pose an additional 64% risk of coronary artery disease in women [110], and this elevated risk is thought to be associated with an increased prevalence of diabetes mellitus and abdominal obesity, especially in young women [111]. VAT in women and IMAT in men are more detrimental to cardiovascular health. In both ageing men and women, SO is independently associated with higher cardiovascular and all-cause mortalities [112,113,114,115,116].

Hypertension is the leading risk factor for CVD. Central obesity is associated with an increased risk of hypertension risk [117]. Young men have a higher prevalence of hypertension compared with age-matched women [118]. Following the onset of menopause, the incidence of hypertension among women begins to surpass that of men [118]. A history of hypertension and chronic obstructive pulmonary disease is more common among women [119,120,121]. In acute aortic dissection, women, at the time of diagnosis, are older than men and complain less frequently of an abrupt onset of pain [120,121]. Obesity is considered an important risk factor for atrial fibrillation (AF), for which the incidence has increased significantly in recent years [122]. A large European study of up to 12.6 years found that the risk of AF is higher in men with a high BMI than in women. The cumulative incidence increased significantly after age 50 in men and after age 60 in women [123]. 

Ischaemic heart disease (IHD), manifested as a myocardial infarction (MI), represents the leading cause of death in women [124]. In patients under the age of 55, the association of hypertension, depression, diabetes, current smoking, a family history of diabetes, and younger age is stronger for MI in women than men [125]. However, a meta-analysis of 1.2 million participants and 95 cohorts showed an increased risk of coronary heart disease in obese individuals without significant sex heterogeneity [126]. In another study, with up to 10 years of follow-up, 899 women from the Women’s Ischaemia Syndrome Evaluation (WISE) prospective cohort (enrolled 1997–2001) were analysed for major adverse cardiovascular events (MACE) and all-cause mortality. It showed that women with overweight and obese signs/symptoms of IHD had a lower risk of long-term all-cause mortality, whereas those with an unfit normal BMI had a higher risk of MACE [127]. 

As a severe manifestation or late stage of various chronic CVD, heart failure (HF) is a sex-heterogeneous disease with different pathophysiology [128]. Based on the left ventricular ejection fraction (LVEF), HF is classified into heart failure with a preserved ejection fraction (HFpEF, LVEF ≥ 50%), heart failure with a reduced ejection fraction (HFrEF, LVEF ≤ 40%), and heart failure with a mildly reduced ejection fraction (HFmrEF, LVEF 41–49%) [129]. One of the most challenging questions for the prevention and management of HF is to differentiate HFpEF from HFrEF, because the dynamic HFrEF treatment strategies may not improve the outcomes of HFpEF [130]. The presence of obesity is strongly associated with traditional heart failure risk factors and increases the subsequent risk of HFpEF [131,132,133,134,135]. The mechanism and clinical outcomes of the HFpEF are also sex-specific [136]. In HFpEF patients, central obesity is more common than general obesity. Energy metabolism abnormalities, haemodynamic disturbances, and cardiac dysfunction affect female patients more than males [137]. An average 34% higher VAT area is quantified by abdominal computed tomography (CT) in women with HFpEF compared with non-HFpEF females, whereas no significantly different VAT area was detected in men with or without HFpEF. A similar trend is also observed in the evaluation of LV filling pressures or pulmonary capillary wedge pressure (PCWP) during exercise in female subjects with a higher VAT area in the same study. Each 100 cm^2^ increase in the VAT area is associated with a 4.0 mmHg higher PCWP [138]. Thus, VAT over-accumulation is sex-specific in the pathophysiology of HFpEF [139]. More pronounced diastolic dysfunction impairment, vascular stiffening, and left ventricular concentric remodelling led to a greater predisposition to HFpEF in women compared with men [140,141,142]. The systolic dysfunction is also significantly more severe in women with diabetes than in the male counterpart [143]. Plasma miRNA-34a, −224, and −452, and microvascular injury marker Angiopoietin-2 are related to diabetic women with HFpEF and left ventricular diastolic dysfunction [144].

## 5. Role of Sex Hormone in Body Fat Distribution and Obesity-Related CVD

Hormonal regulation is thought to be responsible for the difference between men and women. Women differ in their physio-pathological status of cardiovascular function from the reproductive to the perimenopausal and menopausal periods. The female sex hormone, estrogen, plays an important role in the regulation of fat distribution and adipocyte differentiation. Estrogen deficiency in women leads to a significant change in body fat distribution. Estrogen upregulates the number of anti-lipolytic α2A-adrenergic receptors in SAT in young women. Menopause attenuates the lipolytic response, shifting fat accumulation from SAT to VAT [145]. As a result, a postmenopausal female physiologically has a significantly lower level of estrogen than their premenopausal counterparts, resulting in fat re-distribution upon estrogen withdrawal, to a more abdominal obesity phenotype. 

Estrogen protects the female from CVD [146]. In obese women, this protective effect is diminished, even though obesity may lead to a minor increase in estrogen production due to the increased aromatase level from adipocytes [147]. Contrary to the well-known un theory that adipose tissue produces estrogen, studies have shown a negative correlation between BMI and estradiol (a form of estrogen) levels in premenopausal women [148]. In a cohort study, obese and overweight premenopausal females (aged 35 to 47 years old) have a considerably lower level of estrogen than their normal-weighted counterparts [149]. Therefore, the obesity-induced estrogen reduction in premenopausal females reverses the CVD-protection benefit. 

As a gatekeeper of cardiovascular health, endothelial cells (ECs) are demonstrated to balance vasodilation and vasoconstriction and evaluate nitric oxide (NO) bioavailability [150]. The impairment of vasodilation, a hallmark of obesity-induced endothelial dysfunction, is mainly caused by eNOS downregulation and the absence of NO production. To prevent women from experiencing endothelial dysfunction, estrogen and estrogen receptors (ER) are versatile in upregulating NO production by rapid effects and longer-term modulation. Both in vivo and in vitro experiments showed that estrogen mediates the rapid activation of eNOS through the phosphoinositide 3 kinase/protein kinase B (PI-3 kinase/Akt) pathway [151,152,153]. Estrogen also increases the expression of eNOS protein by regulating endothelial eNOS mRNA transcription, thereby achieving long-term and stable protection of the endothelial function [153,154]. In addition to regulating the classical pathway, estrogen has also been found to be involved in regulating endothelial function in other novel pathways. The epithelial sodium channel (ENaC) is shown to be a major determinant of endothelial stiffness. ECs exhibit more prominent stiffness behaviour and a greater ENaC activity in women than in men when NO is lacking. The endothelial SGK-1 activation is involved as a mediator of the link between estrogen signalling and EnNaC activity in women [155]. Moreover, estrogen displayed a regulating behaviour in the expression of the endothelial potassium channels called IK1 and SK3. Reduced IK1 and SK3 expression, together with loss of another vasodilator hydrogen peroxide and elevated superoxide production, contribute to their susceptibility to obesity-related-resistant microvascular dysfunction. The upregulation of IK1 and SK3 caused by the insufficient level of estrogen in obese young females may lead to a lower expression level of potassium channels, which is detrimental to endothelial function. A recent study shows that the deletion of the endothelial cell mineralocorticoid receptor (EC-MR) lowers the IK1 expression and protects only females from susceptibility to obesity and obesity-induced endothelial dysfunction, further proving the negative association between estrogen and CVD risk [156].

Important crosstalk exists between the cell populations of adipose tissue and endothelial cells, whose intercellular crosstalk is critical to the microenvironment [157]. Dysregulated crosstalk between cells within the PVAT is an important contributor to inducing endothelial dysfunction, which is considered as an initiated trigger of atherosclerotic diseases. Unlike in a healthy state, PVAT is substantially different from typical VAT, resulting in adipose tissue dysfunction in obesity. Lesioned PVAT is lipotoxic and can indirectly secrete adipokines and cytokines to impair cardiac and endothelial function [158]. Macrophages are recruited by PVAT to a more inflammatory phenotype, resulting in decreased vasodilation. In addition, macrophage and T lymphocyte infiltration also release inflammatory mediators and chemokines [159]. The increased release of TNF-α and free fatty acids modifies PVAT to a more proinflammatory vasoconstrictor phenotype [160]. Consequently, the dysregulation of adipose tissue endocrine and paracrine signalling, coupled with chronic inflammation, results in dysregulated vascular homeostasis and endothelial dysfunction [161]. Therefore, obesity-induced endothelial dysfunction is a trigger for the initial stage and progression of arteriosclerosis.

Oral contraceptives in premenopausal women also reduce the estrogen peak during the female menstrual period, which may have deleterious effects on cardiovascular and metabolic outcomes. The cessation of estrogen production either in menopausal women or premenopausal women on oral contraception abolishes this beneficial effect of estrogen and predisposes women to hypertension [162]. Clinical studies have shown a positive correlation between the use of oral contraceptives and the development of the metabolic syndrome. Long-term use of oral contraceptives impairs the protective effect of estrogen and increases the incidence of hypertension in young women [163]. For example, ethinyloestradiol (EE)-containing oral contraceptives have the biological potency to activate the RAAS by increasing the hepatic production of angiotensinogen, the pre-cursor for angiotensin-2, which is noted for the elevation of blood pressure [164]. EE also causes endothelial dysfunction and a reduction in nitric oxide production by directly acting and causing alterations to the vascular wall. Oral contraceptive use-associated vascular impairment, in addition to the widely reported EE-induced changes in the coagulation system because of increased thrombin activity and a reduction in endogenous anticoagulants, increases the risk of cerebrovascular incidents such as strokes [165,166]. In addition, many studies have reported that oral contraceptives increase triglyceride levels and negatively impact the profile of lipids such as total cholesterol, low-density lipoproteins, and high-density lipoproteins with accompanying risks of cardiovascular diseases [163]. The side-effect of such kind of female oral contraception on cardiovascular health has a great implication for the innovation of hormonal male contraceptives as being reasonably safer with no significant cardiovascular risks, however, relatively fewer males use hormonal contraceptives, and the study durations were much shorter [167]. Males also have a relatively stable estrogen level compared to females. The stability of hormonal regulation puts obese males at the advantage of suffering from a less severe increase in blood pressure compared to females who have similar body weight gain, especially postmenopausal ones. As a result of hormonal fluctuation, obese female mice develop more prominent reduced insulin sensitivity, diastolic dysfunction, interstitial fibrosis, cardiac stiffness, altered Ca2+ handling, and a marked decrease in Akt/eNOS activation earlier than their male counterparts [168]. 

Not only does adipose tissue produce estrogen, but it also synthesises testosterone. Excessive adipose tissue overexpresses aromatase which could catalyse testosterone to estrogen. The decreased amount of testosterone circulating in obese men contributes to augmenting cardiovascular risk [169,170]. Small-scale clinical studies have shown that endogenous testosterone has a protective effect on endothelial cells [171,172]. By regulating the expression and activity of eNOS, testosterone deficiency may contribute to endothelial dysfunction and atherosclerosis by reducing NO levels [173]. In addition, testosterone may affect endothelial repairment by regulating the proliferation and migration of EPCs [174,175]. However, multiple studies have not reached effective or consistent conclusions on the application of testosterone replacement therapy in CVD, and longer-term follow-up and the inclusion of more studies may be required [176].

Overall, the above evidence shows a more severe cardiovascular impact of sex hormones on obese females than male counterparts, indicating the use of hormone therapy as a potentially viable way for both females and males to reduce the development of obesity and provide vascular protection.

## 6. Role of Adipokines in Causing Obesity-Related CVD

Another factor contributing to obesity-related CVD risk is the disrupted production of adipokines, adipose tissue cytokines that have both anti- and pro-inflammatory functions. Some adipokines have emerged as pro-inflammatory factors that ultimately worsen CVD. In a normal-weighted state, there is a balance between pro- and anti-inflammatory factors for homeostasis [177]. Obesity alters the adipose tissue microenvironment to a more pro-inflammatory state with the production of inflammatory factors and the recruitment of inflammatory cells. Consequently, these pro-inflammatory factors act as mediators between inflammation and chronic diseases, contributing to long-term chronic inflammation, diminishing immunity in the body, and leading to several metabolic diseases [178]. Metabolic disorders act as disturbances of the body functions and are associated with potentially life-threatening diseases such as cardiovascular diseases [179]. Among all discovered adipokines, leptin, lipocalin-2, visfatin, and resistin are the four most intensively studied, and are shown to exert pro-inflammatory properties [180]. The common characteristic of these adipokines is that they are all positively related to obesity and obesity-related complications such as insulin resistance and CVD, potentially leading to obesity-related comorbidities [181]. 

The production of leptin, one of the most well-studied adipokines in CVD, is significantly increased in obese patients, and the mechanisms by which it contributes to the development of hypertension show significant sex differences in young patients [181]. Specifically, in young obese men, leptin-derived sympathetic activation promotes hypertension, whereas premenopausal women develop hypertension through extra-sympathetic mechanisms [182,183]. These sex-based differences disappear in older patients. For postmenopausal women, obesity-induced increases in leptin levels induce endothelial dysfunction and hypertension through sympathetic activation, which is one of the non-negligible triggers for the occurrence of cardiovascular accidents in women. Moreover, lipocalin dysregulation and leptin downregulation promote insulin resistance development in premenopausal and postmenopausal women [145,184]. The prevention and treatment of insulin resistance are, therefore, essential to prevent obese women from developing CVD.

Another well-studied pro-inflammatory adipokine called lipocalin-2 showed sex-specific differences in circulating levels and association with obesity and obesity-related cardiometabolic malfunctions [185]. This adipokine is primarily produced by adipocytes, visceral cells (liver, lung, and kidney cells) [186], and immune cells (neutrophils and macrophages) [187]. It is known to have an effect on CVD such as atherosclerosis [188], endothelial inflammation [189], and cardiac remodelling [190]. The most likely pathway associated with cardiovascular consequences are inflammatory pathways, such as binding to matrix metalloproteinase 9 (MMP9) and activating downstream inflammation [191]. One small cohort study (*n* = 16) has shown that overweight males have higher levels of serum lipocalin-2 (4.4 ± 2.3 nmol/L) than their female counterparts (3.8 ± 2.3 nmol/L) [192]. A larger cohort study containing 229 obese subjects complemented the previous smaller study, as it was also observed that obese males have higher serum levels of lipocalin-2 (117.7 mg/L) compared to obese females (92.9 mg/L) [193]. Current research using both animal and human participants suggests that obesity may increase heart size, which is partially mediated by an increase of lipocalin-2 expression and thus leads to heart failures and various other cardiovascular complications [190]. This evidence suggests that obese males who have a high serum lipocalin-2 level are more susceptible to developing CVD. Therefore, lipocalin can be an important biomarker of CVD, especially for obese patients.

Visfatin is an adipocytokine mainly produced by macrophages and adipocytes in visceral fat, although the exact site of visfatin production is still under investigation [194,195]. The main role of visfatin is regulating the glycaemic level [180]. Visfatin mimics insulin, binding insulin receptors to downregulate the glucose level, damages beta cell function, and potentially fluctuates insulin sensitivity [194]. Studies have shown that circulating the visfatin level is higher in obese individuals, especially those with type II diabetes and susceptible to CVD [195,196,197], when compared to their counterparts. Studies have also shown the effect of an increased level of visfatin in endothelial dysfunction and vascular inflammation which may lead to CVD [198,199]. For example, the association between high serum and immune cell-expressed visfatin and the increased risk of atherosclerosis and the formation of plaques has been well-established [187,200]. A cohort study involving examining carotid plaques isolated from two groups (experienced symptoms of CVD versus non-symptomatic patients) of a total of 21 patients has concluded that high levels of macrophage-expressed visfatin play a role in destabilising the plaques and lead to severe complications of arteriosclerosis, coronary disease, and myocardial infarction [187]. Moreover, visfatin has a proliferative effect on cardio-fibroblasts, which increases myocardial fibrillation [187,201]. Taken together, visfatin has been proven to preferentially express in the visceral fat [199,202]; hence obese males may be more vulnerable to the upregulation of visfatin and the consequences of such a change to cardiovascular function than obese females due to the higher visceral adiposity in males than in females. Other sex differences in the visfatin level are controversial and remain to be elucidated.

Pro-inflammatory resistin is mainly produced by adipocytes, mononucleated immune cells, and lymphoid organs such as the spleen. Its elevation is associated with overweight and obesity and can interfere with insulin action [203]. The increased level of serum resistin competes with other immune molecules to bind to Toll-like receptors 4 (TLR4) and Adenylyl Cyclase Associated Protein 1 (ACAP1), and, therefore, activate various downstream inflammatory pathways. The activation of such pathways may result in cardiovascular damage due to vascular inflammation and plaque detachment [204]. For example, a cohort study of 220 cases of CVD or suspected CVD indicated that a high serum level of resistin could be a biomarker of CVD as it has shown to have a negative effect in acute coronary syndrome [205]. Although the sex difference in human resistin levels has not yet been well-characterised, several studies have proven that resistin is regulated in a sex-dependent manner. An in vivo study showed that the level of adipose tissue-expressed resistin increased as mice gained weight, with female mice having a higher level of resistin than male mice at all ages, and only the cessation of female sex hormones would change the resistin level, which is an indication of the sex-specific regulation of resistin [206]. A human clinical study of 213 obese patients (divided into two groups according to their sex) revealed that increased serum resistin levels were only associated with an increased risk of CVD in females [207]. This evidence concludes that obese females with elevated resistin levels are more susceptible to CVD than their male counterparts. Resistin might, thus, be a new biomarker for obesity in females.

Understanding these pro-inflammatory adipokines will significantly facilitate the development of sex-based personalised medicine. The concepts of leptin therapy are relatively mature. Several clinical studies are going on using leptin and leptin agonists and sensitisers as treatment agents [208]. For example, obese patients, especially females, showed increased levels of serum leptin as a risk marker of obesity, indicating that leptin is not utilised efficiently to regulate body weight, a phenomenon called “leptin resistance”. Therefore, an engineered and recombined leptin dose [209], leptin receptor agonists, and leptin sensitisers [210] have been developed to reduce this phenomenon and make more effective leptin molecules across the blood–brain barrier to reach its effective site to treat obesity and obesity-related CVD [208], typically for treating premenopausal obese females. Similarly, monoclonal antibodies targeting lipocalin-2 have been intensively proposed as a cancer treatment [211]. As research has already proved the association between lipocalin-2 and obesity and obesity-related CVD, monoclonal anti-lipocalin-2 could be a feasible treatment option. Although the specific therapies targeting visfatin and resistin have not yet been established, recent findings have found an anti-inflammatory factor called “adropin” and have been speculated as the antagonist to suppress the mRNA level of these two adipokines [212,213] which might become a potential therapeutic strategy to reduce obesity and obesity-related CVD in both males and females.

In summary, the above-mentioned pro-inflammatory adipokines possess a negative effect on CVD in a sex-dependent way. As personalised medicine develops, clinical medicine is shifting toward a more customised disease prediction, diagnosis, and treatment for CVD [214]. Therefore, information about sex differences in obesity and obesity-related CVD would provide more insights into the potential biomarkers of CVD diagnosis and new sex-specific therapies to treat CVD of obese patients through the development of personalised and sex medicine. In the future, antibodies, antagonists, or pro-inflammatory-specific miRNA therapies that regulate the bioavailability and bioactivity of these adipokines would be useful in treating obese CVD male and female patients, respectively [209].

## 7. Genetic Regulation of Sexual Dimorphism in Adiposity

Unlike infectious diseases, complex environmental and genetic factors are the root of chronic non-communicable diseases [215]. In a nuclear family study, the maximum heritability of fat is estimated to be 46–60% whereas the range of fat distribution is 29–48%, which has a null correlation with overall obesity. This result indicates that the heritability of fat is greater than it of fat distribution. In addition, the significant correlation of parent–offspring and sibling correlation patterns rather than the usual spousal patterns suggests that genes play a role in explaining at least part of heritability [216]. In addition, the body fat distribution has a strongly heritable trait emphasising the genetic regulation of sexual dimorphism in the adiposity [217,218,219,220,221]. Genetic loci located in the sex chromosomes and the autosomal genome are crucial for sexual dimorphism [222,223,224,225]. Anthropometric traits rather than BMI, weight, and height are the key for genetic association studies to elucidate the genomic architecture of obesity in humans. Shungin D et al. conducted genome-wide association (GWAS) meta-analyses of traits related to WHR in up to 224,459 individuals and identified 20 loci with significant sexual dimorphism. The loci are enriched for genes expressed in adipose tissue and implicated in adipogenesis, angiogenesis, transcriptional regulation, and insulin resistance [226]. While some heritable traits linked to BMI, height, and weight are similar, other genetic architectures related to WC and WHR are quite distinct between males and females [218,220]. The loci for BMI variations do not show sex differences, but suggest a role of the central nervous system in obesity susceptibility [227]. By contrast, the heritability of WHR is significantly larger in women than in men [226]. Lotta and colleagues used genetic scoring to compare the different contributions of lower gluteofemoral fat and higher abdominal fat to the cardiometabolic consequences of people with higher WHR. The test found that 202 independent genetic variants were associated with BMI-adjusted WHR. The waist specificity score is related to higher VAT and SAT, but no relationship with hip or leg fat. In contrast, the hip-specific score was associated with lower-hip and extremity fat as well as lower SAT, but not significantly correlated with VAT. Both scores increase the risk of poor metabolic characteristics, type 2 diabetes, and coronary artery disease [228,229]. In addition, a study based on DEXA and bioelectrical impedance measurements estimates that a genetically determined increase in VAT may be a powerful and independent determinant of cardiovascular and metabolic diseases. Studies have found that higher estimated VAT is associated with an increased risk of hypertension, cardiovascular events, type 2 diabetes, and hyperlipidaemia [230].

A GWAS analysis ancestry from the Framingham Heart Study (FHS) and the Multi-Ethnic Study of Atherosclerosis (MESA) screened a unique locus near *TRIB2* and loci at *TCF21*, strongly associated with PCAT but independent of VAT and total fat mass. These emerging shreds of evidence emphasised the concept that ectopic fat distribution has a unique genetic basis [231]. Additionally, a locus near *THNSL2* and *FABP1* has been found for VAT in women but not in men [232]. Similarly, variants near *TFAP2B* seem to influence central obesity through their effects on overall obesity/fat mass, while *LYPLAL1* shows a strong female correlation with the fat distribution [233]. Four previously constituted loci and three new anthropometric characteristic loci (namely *GRB14/COBLL1*, *LYPLAL1/SLC30A10*, *VEGFA*, *ADAMTS9*, *MAP3K1*, *HSD17B4*, and *PPARG*) all have a full genome in females, but significantly, not in males [234]. It has been confirmed that *GRB14* is highly related to fasting insulin, triglycerides, and high-density lipoprotein in American-African women, but not in men. These data also indicated the concept that a body fat distribution site is not related to systemic obesity [235]. GWAS-combined MRI data enforced this conclusion and carrying a more favourable allele for obesity is associated with a lower waist circumference but higher hip circumference in women, while the genetic score favourable for obesity is associated with lower liver fat in women, but not with fatty liver in men. Furthermore, the difference is not statistically significant, and the association between favourable obesity alleles and lower liver fat in premenopausal women is twice that of postmenopausal ones, indicating that sex hormones may have a certain regulatory role in the expression of favourable obesity alleles [236]. In addition to assessing the contribution of sex chromosomes, the stratification of the GWAS results by sex is a valuable approach that reveals the gender dichotomy of visceral obesity [237]. The GWAS for visceral adiposity (assessed as a waist-to-hip ratio adjusted for the body mass index) identifies an autosomal locus with significant sex-specific effects. Some loci associated with visceral obesity were sex-specific and also present in both sexes, but had opposite effects on the waist-to-hip ratio [238]. Another study performed shows that the *Lyplal1* gene specifically counteracts fat accumulation in female mice caused by a high-fat diet [239].

Overall, the familial contribution to fat distribution is stronger in women as compared to men [240]. The sexual dimorphism in the heritability suggests that biological pathways are uniquely, specifically, or differentially involved in the determination of body fat distribution.

## 8. Summary

Obesity is an important risk factor for cardiovascular diseases. Heterogeneity in the regional deposition of fat is more deleterious than total body adiposity. Many studies show associations between cardiovascular risk factors and directly-measured VAT which are stronger than those observed with typical anthropometric measures. The distribution of body fat differs between men and women. While the former is associated with increased cardiovascular risk, the latter, on the other hand, has more SAT and BAT. The female pattern of fat distribution is associated with improved cardiovascular risk at a similar BMI. However, ectopic fat deposition within the abdomen, pericardium, and neck is more strongly associated with women’s adverse cardiovascular risk than men. Female fat distribution and expression regulation may be more genetically affected than males by environmental factors. The molecular mechanism of this sex dimorphism may be beyond the modulation of sex hormones.

## Figures and Tables

**Figure 1 ijms-23-09338-f001:**
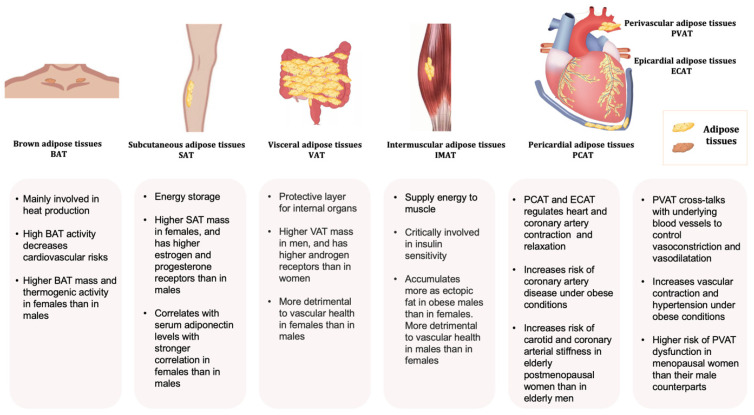
Typical adipose tissues, main functions, and some sex-related differences.

## Data Availability

Not applicable.

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
