# Peer review of "Sex Differences in Adiposity and Cardiovascular Diseases"

_ijms, 2022, doi:10.3390/ijms23169338_

Round 1

Reviewer 1 Report

Review article by Haoyun Li et al. entitled ‘Sex Differences in Adiposity and Cardiovascular Diseases’ is about how the different types of fat and different location of fat deposition affect the metabolic system differently in male and female and leads to CVD and implicated in other diseases. This review article deals with a very important subject of current research area.

After careful review I found this review article is well written well as qualifies the journal criteria.

However, to reach a decision I have some comments and questions to be clarified by the authors.

Please correct sentence in line 224………. significant change find n body the conditions that more prone to…..

Line 230-231 seems contradictory.. ‘even though obesity may further lead to a minor increase in estrogen production [121]’ Please clarify .. This condition is due to abdominal fat accumulation or general? Because this is already discussed later in the review article.

Line 282-289 Authors have given many web links which is very confusing because all the links always bring to same journal paper specially in reference section…… Please correct that either put reference instead of web links. Why have authors given all the web link from the same paper? please clarify. Whether authors add citation of that paper in the reference section?

Web Link article…Andrea Rodrigues Sabbatini and Georgios Kararigas corresponding ‘Estrogen-related mechanisms in sex differences of hypertension and target organ damage’ Biol Sex Diffe 2020.

Endothelial dysfunction is an important pathway of CVD it would be very helpful for readers if authors add some sentences related to adipose tissue or fat metabolism in endothelial cells in male and female and their consequences. This is my opinion if authors are not willing to do it, it will not affect the decision.

Author Response

Response to Reviewer 1 Comments

We would like to thank the reviewers and editors for their time and valuable feedback. We have now extensively revised the manuscript to address all the comments from the expert reviewers. The detailed point-to-point responses are provided below.

Point 1: Please correct sentence in line 224…        significant change find n body the conditions

that more prone to…..

Response 1: We thank the reviewer’s suggestion. The sentence in line 224 have been corrected accordingly.

Point 2: Line 230-231 seems contradictory.. ‘even though obesity may further lead to a minor increase in estrogen production [121]’ Please clarify .. This condition is due to abdominal fat accumulation or general? Because this is already discussed later in the review article.

Response 2: We agree with the reviewer’s comment. After careful checking of the reference, we have clarified that “in obese women, this protective effect is diminished, even though obesity may further lead to a minor increase in estrogen production due to the increased aromatase level from abdominal adipocytesand revised in the review accordingly.

Point 3: Line 282-289 Authors have given many web links which is very confusing because all the links always bring to same journal paper specially in reference section…… Please correct that either put reference instead of web links. Why have authors given all the web link from the same paper? please clarify. Whether authors add citation of that paper in the reference section?

Response 3: We thank the reviewer’s s comment. This reference discussed an important issue that estrogen as a mediator leading to sex differences in hypertension. The reason we intend to include this topic because its discussion is comprehensive and related to sexual dimorphism and obesity-related cardiovascular abnormalities closely.

With the gratitude for reviewer’s kind reminder, the duplicated web links which caused by the system error were removed thoroughly.

Point 4: Endothelial dysfunction is an important pathway of CVD it would be very helpful for readers if authors add some sentences related to adipose tissue or fat metabolism in endothelial cells in male and female and their consequences. This is my opinion if authors are not willing to do it, it will not affect the decision.

Response 4: We thank the reviewer’s suggestion. Endothelial dysfunction is a crucial subtopic which should be discussed in the obesity-related disease. We devoted a paragraph to the 4th subtopic : Sexual dimorphism and obesity-related cardiovascular abnormalities, to address this issue accordingly.

Reviewer 2 Report

Dr. Li and colleagues present a review focusing on the current knowledge of the sex differences related to adiposity and associated cardiovascular complications.

I think that some points could be explored. Different cardiovascular diseases have been associated with obesity and adipose tissue, such as atrial fibrillation and heart failure with preserved ejection fraction (HFpEF), and others.

For example, HFpEF is more prevalent in women. Moreover, HFpEF is associated with obesity. I think that you can discuss this, and include subtopics related to specific cardiovascular diseases.

The adipokines have emerged as pro-inflammatory factors (visfatin, resistin…). There are sex differences between these adipokines? How can be associated with cardiovascular diseases?  Please discuss how this information could be helpful in clinical practice and how you can implement this.

Please revise the English, and some formatting errors (example line 61). The references on page 6 should be clarified and well formatted. Figure 2 should be included in the paper. 

Author Response

Response to Reviewer 2 Comments

We would like to thank the reviewers and editors for their time and valuable feedback. We have now extensively revised the manuscript to address all the comments from the expert reviewers. The detailed point-to-point responses are provided below.

Point 1: I think that some points could be explored. Different cardiovascular diseases have been associated with obesity and adipose tissue, such as atrial fibrillation and heart failure with preserved ejection fraction (HFpEF), and others.

Response 1: We thank the reviewer’s suggestion. The discussion of different obesity-related cardiovascular diseases( HFpEF, Atrial fibrillation, Myocardial infarction, and Hypertension) has been included in the 4th subtopic : Sexual dimorphism and obesity-related cardiovascular abnormalities. It more profound echoes of the theme to make these changes.

Point 2: The adipokines have emerged as pro-inflammatory factors (visfatin, resistin…). There are sex differences between these adipokines? How can be associated with cardiovascular diseases? Please discuss how this information could be helpful in clinical practice and how you can implement this.

Response 2: We thank the reviewer’s suggestion.The related information has been expanded in the 6th subtopic : Role of adipokines in causing obesity-related CVD. Four pro-inflammatory adipokines namely leptin, lipocalin-2, visfatin and resistin that are experimentally proven to exert sex-difference are included. The applicability and possible approaches of implementation in clinical practice for cardiovascular diseases were also addressed.

Point 3: Please revise the English, and some formatting errors (example line 61). The references on page 6 should be clarified and well formatted. Figure 2 should be included in the paper.

Response 3: We thank the reviewer’s s comment. Correction of grammatical constructions and format revision was performed as the reviewers indicated accordingly.

However, after the major revision and careful discussion, all the authors concluded that to include the Figure 2 from the cited reference may need a second thought. It may possibly over- address the position of estrogen than other influencing factors of sexual dimorphism and obesity- related cardiovascular diseases. We sincerely thank the professional suggestions above and enlightening new insights into this review.